# Analysis of SARS-CoV-2 vertical transmission during pregnancy

Claudio Fenizia [1,12], Mara Biasin [2,12], Irene Cetin [3], Patrizia Vergani[4], Davide Mileto[5], Arsenio Spinillo[6], Maria Rita Gismondo[5], Francesca Perotti [7], Clelia Callegari[4], Alessandro Mancon [5], Selene Cammarata [8], Ilaria Beretta[9], Manuela Nebuloni [10], Daria Trabattoni [2], Mario Clerici[1,11] & Valeria Savasi [8✉]

The impact of SARS-CoV-2 infection during gestation remains unclear. Here, we analyse the viral genome on maternal and newborns nasopharyngeal swabs, vaginal swabs, maternal and umbilical cord plasma, placenta and umbilical cord biopsies, amniotic fluids and milk from 31 mothers with SARS-CoV-2 infection. In addition, we also test specific anti-SARS-CoV-2 antibodies and expression of genes involved in inflammatory responses in placentas, and in maternal and umbilical cord plasma. We detect SARS-CoV-2 genome in one umbilical cord blood and in two at-term placentas, in one vaginal mucosa and in one milk specimen. Furthermore, we report the presence of specific anti-SARS-CoV-2 IgM and IgG antibodies in one umbilical cord blood and in one milk specimen. Finally, in the three documented cases of vertical transmission, SARS-CoV-2 infection was accompanied by a strong inflammatory response. Together, these data support the hypothesis that in utero SARS-CoV-2 vertical transmission, while low, is possible. These results might help defining proper obstetric management of COVID-19 pregnant women, or putative indications for mode and timing of delivery.

[1] Department of Pathophysiology and Transplantation, University of Milan, Milan, Italy. [2] Department of Biomedical and Clinical Sciences, University of Milan, Milan, Italy. [3] Department of Woman, Mother and Neonate Buzzi Children's Hospital, ASST Fatebenefratelli-Sacco, Department of Biomedical and Clinical Sciences, Milan, Italy. [4] Department of Maternal Fetal Medicine, Fondazione MBBM, San Gerardo Hospital, University of Milan-Bicocca, Monza, Italy. [5] Clinical Microbiology, Virology and Bio-emergence Diagnosis, ASST Fatebenefratelli-Sacco, Department of Biomedical and Clinical Sciences, University of Milan, Milan, Italy. [6] Department of Obstetrics and Gynecology, IRCCS Fondazione Policlinico San Matteo, University of Pavia, Pavia, Italy. [7] Department of Obstetrics and Gynecology, IRCCS Fondazione Policlinico San Matteo, Pavia, Italy. [8] Unit of Obstetrics and Gynecology, ASST Fatebenefratelli-Sacco, Department of Biological and Clinical Sciences, University of Milan, Milan, Italy. [9] Division of Infectious Diseases, San Gerardo Hospital, ASST Monza, Monza, Italy. [10] Pathology Unit, ASST Fatebenfretalli-Sacco, Department of Biological and Clinical Sciences, University of Milan, Milan, Italy. [11] IRCCS Fondazione don Carlo Gnocchi, Milan, Italy. [12] The authors contributed equally: Claudio Fenizia, Mara Biasin. ✉email: valeria.savasi@unimi.it

The novel coronavirus disease 2019 (COVID-19) pandemic is currently spreading worldwide. The number of confirmed cases currently exceeds 11.5 million people, with approximately 590,000 deaths, and Italy represents one of the most affected countries[1–3]. Severe COVID-19 cases exhibit a dysfunctional immune response characterized by higher blood plasma levels of interleukin (IL)-1β, IL-2, IL-6, IL-7, IL-10, granulocyte colony-stimulating factor, interferon gamma-induced protein-10 (C-X-C motif chemokine ligand 10 (CXCL10)), monocyte chemoattractant protein-1 (C-C motif chemokine ligand 2 (CCL2)), macrophage inflammatory protein 1α (CCL3) and tumour necrosis factor, which mediate widespread lung inflammation and fail to successfully eradicate the pathogen[1,4–7].

Maternal physiological adaptations to pregnancy are known to increase the risk of developing severe illness in response to viral infections, such as influenza; preliminary data suggest that the prognosis of severe acute respiratory syndrome coronavirus 2 (SARS-CoV-2) infection could also be more severe in pregnant women[8]. Vertical transmission of SARS-CoV and Middle East respiratory syndrome (MERS), the two other animal coronaviruses known to infect humans, was never documented to occur. However, the number of reported cases of infected pregnant women was very low and not sufficient to draw firm conclusions (12 reported cases for SARS-CoV and 11 for MERS)[9,10]. Conversely, as the number of SARS-CoV-2-positive patients is rising worldwide, multiple reports focus on SARS-CoV-2-positive pregnant women[11–16]. No trace of the virus was detected by real-time PCR[11,12,14,15,17]; however, two independent manuscripts described elevated SARS-CoV-2-specific immunoglobulin G (IgG) and IgM antibody levels in the blood of three newborns of SARS-CoV-2-infected mothers[18,19]. As IgG, but not IgM, are normally transferred across the placenta, this is suggestive of in utero infection[18,19]. Moreover, placental submembrane and cotyledon were reported to be positive for the virus in a 20-week miscarriage of a SARS-CoV-2-positive pregnant woman[20].

As recently reported, the two known SARS-CoV-2 receptors angiotensin-converting enzyme 2 (ACE2) and transmembrane protease serine 2 are widely spread in specific cell types of the maternal–foetal interface[21]. Therefore, the impact of the virus on placenta and the potential for the vertical transmission of SARS-CoV-2 need to be further carefully addressed.

Here, we report the presence of the SARS-CoV-2 genome in umbilical cord blood and in at-term placentas, in vaginal mucosa of pregnant women and in milk specimen. Furthermore, we report the presence of specific anti-SARS-CoV-2 IgM and IgG antibodies in the umbilical cord blood of pregnant women, as well as in one milk specimen. Finally, an intense inflammatory response is triggered by SARS-CoV-2 infection in pregnant women at both the systemic and placental levels and, concerningly, in umbilical cord blood plasma. Taken together, these results suggest that, although rare, SARS-CoV-2 in utero vertical transmission is possible and that the well-known SARS-CoV-2-related inflammatory status is extended to foetuses. Understanding the biological behaviour of the virus during pregnancy is essential for defining proper obstetric management of pregnant women with COVID-19.

## Results

**Population**. Four patients were classified as severe cases, defined by the need for urgent delivery for the deterioration of maternal conditions or by intensive care unit (ICU)/sub-intensive care admission. A radiological confirmation of interstitial pneumonia was obtained on admission or antepartum for all the severe cases and in 10 (32%) of the mild cases. Pharmacological treatment during the antepartum period of hospitalization is reported in

**Table 1 Baseline characteristics of the study population on admission.**

| Characteristics at delivery | Total study population (n = 31) |
|---|---|
| Maternal baseline characteristics | |
| Maternal age, years, median (range) | 30 (15–45) |
| RT-PCR assay of a maternal nasopharyngeal swab | |
| Positive, n (%) | 30 (97) |
| Negative, n (%) | 1[a] (3) |
| Prepregnancy BMI, kg/m², median (range) | 23 (17–37) |
| Known sick contact, n (%) | 10 (32) |
| Smoking habit, n (%) | 0 |
| Ethnicity, Caucasian, n (%) | 21 (68) |
| Chronic comorbidity, n (%) | 11 (35) |
| Parity, nulliparous n (%) | 14 (45) |
| Flu vaccination in pregnancy, n (%) | 8 (26) |
| Ante-partum therapy | |
| Antibiotic, n (%) | 10 (32) |
| Lopinavir/Ritonavir, n (%) | 8 (26) |
| Hydroxychloroquine, n (%) | 13 (42) |
| Oxygen support without ICU admission, n (%) | 4 (13) |
| Positive chest X-ray, n (%) | 14 (45) |
| Severe case, n (%) | 4 (13) |
| Admission to ICU, n (%) | 1 (3) |

[a]Subject no. 31, SARS-CoV-2 recovered at the time of enrolment.

Table 1. In the only severe case of preterm labour (subject no. 17), corticosteroids for respiratory distress syndrome prophylaxis were administered.

Maternal and pregnancy outcomes in the study population are reported in Table 2. Regarding the mode of delivery, three patients underwent emergency delivery for maternal respiratory indication. Among the severe cases, one needed postpartum admission to the ICU and invasive ventilation for 11 days in total.

Subject no. 31 became negative at week 35 of pregnancy and delivered spontaneously at week 38. Except in one case (subject no. 17), all pregnancies were full term. Subject no. 17 was admitted preterm at 33 + 6 weeks with fever and dyspnoea and delivered spontaneously at 34 + 4 weeks. A female baby was born, weighing 2180 g, with an Apgar score of 9 and 10 at 1 and 5 min, respectively, with a pH of umbilical artery of 7.14. The newborn was diagnosed with SARS-CoV-2 infection through a nasopharyngeal swab and was admitted to the neonatal ICU for prematurity. Subject no. 25 spontaneously delivered at week 39 + 2. A male baby was born, weighing 3340 g, with an Apgar score of 9 and 10 at 1 and 5 min, respectively, and the umbilical artery pH was 7.14. The newborn was diagnosed with SARS-CoV-2 infection through a nasopharyngeal swab upon delivery, while he tested negative 48 h later. Except for the two abovementioned cases, no other newborns were positive for SARS-CoV-2 genome detection by nasopharyngeal swabs. Except for two cases, all newborns were breastfed. All the neonates were healthy and the two SARS-CoV-2-positive babies were totally asymptomatic.

**Viral genome and antibody detection**. We investigated the presence of SARS-CoV-2 in the collected specimens, as shown in Table 3 and summarized in Table 4a. We detected the SARS-CoV-2 genome in 2 (6%) maternal plasma samples (subject nos. 4 and 17), both of them characterized by a severe clinical outcome. Moreover, we detected the presence of the SARS-CoV-2 genome in vaginal swabs, placental tissue and cord plasma from subject no. 17. Moreover, we detected SARS-CoV-2 RNA in placental

**Table 2 Maternal and pregnancy outcomes in the study population.**

| | Total study population $n = 31$ |
|---|---|
| Delivery mode | |
| Vaginal, n (%) | 25 (81) |
| Caesarean section, n (%) | 6 (19) |
| GA at delivery, weeks median (range) | 39 (34.4–41.4) |
| Induction of delivery related to COVID-19, n (%) | 6 (19) |
| Caesarean section for severe maternal illness related to COVID-19, n (%) | 3 (9) |
| Preterm delivery, n (%) | 1 (3) |
| Foetal gender, male, n (%) | 18 (58) |
| Birth weight, g, median (range) | 3200 (2180–4165) |
| Umbilical artery pH, median (range) | 7.31 (7.14–7.43) |
| APGAR score 5' <7, n (%) | 1 (3) |
| Infected neonates, positive, n (%) | 2 (6) |
| NICU admission, n (%) | 2 (6) |
| Skin to skin, n (%) | 4 (13) |
| Breastfeeding, n (%) | 29 (94) |

tissue from subject no. 25. The newborn of subject no. 17 could be classified as having a confirmed congenital infection because of the detection of viral genome by PCR in a nasopharyngeal swab at birth (collected after the baby was cleaned), in placental sample and in umbilical cord plasma. Subject no. 25 could be classified as a possible neonatal infection acquired intrapartum because of the detection of viral RNA by PCR in nasopharyngeal swabs at birth (collected after the baby was cleaned) but not at 24-48 h of age and of SARS-CoV-2-specific antibodies in the umbilical cord plasma.

We detected the SARS-CoV-2 genome in one milk specimen only from a severe clinical outcome case (subject no. 1). Neither the 3 tested amniotic fluids nor the 12 umbilical cords were positive (Tables 3 and 4a).

SARS-CoV-2-specific IgM were detected in 32% of the maternal plasma, while virus-specific IgG were present in 63% of cases. Interestingly, we detected the presence of IgM in the cord plasma in one newborn only (no. 25), whose placenta tested positive for SARS-CoV-2, while IgG were present in 40% of the umbilical cord plasma. Subject no. 1 displayed IgM in milk sample (Table 3, summarized in Table 4b).

**Inflammatory response in SARS-CoV-2-positive subjects.** To determine whether SARS-CoV-2 infection results in an alteration of inflammatory gene expression in placental tissue, we analysed the expression of 84 genes involved in the inflammatory response in 4 selected placental biopsies. The results showed that placentas from SARS-CoV-2-infected patients (subject nos. 17 and 25) display a generalized immune activation profile compared to the uninfected profile (CTR−) (Fig. 1). Likewise, subject 31, who was infected at gestational week 32 but fully recovered 4 weeks before delivery, showed an increased inflammatory profile when compared to CTR−. Notably, such hyperactivation status was far more evident in a placental biopsy from subject no. 17 and even more in one from no. 25, whose nasopharyngeal swab tested positive for SARS-CoV-2 genome detection at T1, compared to the placenta biopsy from patient 31, who tested negative at the time of delivery. The genes whose mRNA expression was clearly upregulated in subjects 17 and 25 are involved in different aspects of the inflammatory response and include effector cytokines and chemokines, adaptive immunity mediators, downstream signal-ling molecules and Toll-like receptors (Fig. 1).

An in-depth investigation of the cytokine/chemokine profile was carried out next in CTR−, 17, 25 and 31 subjects due to their peculiarities. A 27-cytokine multiplex assay was performed on plasma isolated from maternal and funicular blood samples. Overall, the results obtained on maternal plasma confirmed what was observed at the mRNA level in the placentas. Briefly, proinflammatory antiviral cytokines and chemokines were upregulated in patient nos. 17 and 25, compared to subjects CTR− and 31. Subject 25 displayed a more pronounced proinflammatory profile. In particular, IL-1β and IL-6 production was higher in subject 25 compared to all the other subjects (Fig. 2a). The same analysis was performed on funicular plasma. As for maternal plasma, the concentration of proinflammatory molecules was strongly increased in newborns from subject nos. 17 and 25. This increase was mostly evident for the chemokines IL-8, CCL2, CCL3, CCL5 and CXCL10 (Fig. 2b).

## Discussion

We report that the SARS-CoV-2 genome is found in umbilical cord blood[11,19,22–26]. Additionally, consistent with the literature, we found the SARS-CoV-2 genome in the vaginal mucosa of a pregnant woman[23,27] and in at-term placentas[20,27–31]. Further-more, we report the presence of specific anti-SARS-CoV-2 IgM and IgG antibodies in the umbilical cord blood of pregnant women, as well as in milk specimens. Notably, we also provide evidence that SARS-CoV-2 RNA can be found in milk specimens. Our data indicate that in utero vertical transmission is possible in SARS-CoV-2-positive pregnant women, consistent with previous reports[27]. Finally, this report describes the inflammatory response triggered by SARS-CoV-2 infection in pregnant women both at the systemic and placental levels.

Our results suggest in utero vertical transmission in 2 of the 31 (6%) enrolled SARS-CoV-2-positive women. One case was characterized by a severe clinical outcome (subject no. 17), while the other case was classified as mild (subject no. 25); we speculate that the risk of mother-to-child viral transmission does not directly depend on the severity of disease progression. Supporting this observation, the clinical history as well as the results of the viro-immunological test performed on these two subjects were divergent. Subject no. 17, characterized by severe conditions, was SARS-CoV-2 positive in different specimens, including maternal plasma, vagina as well as umbilical cord plasma and placenta. In this case, we hypothesize that the virus spreads around the body through the bloodstream, reaching the vagina and the placenta and finally infecting the foetus. Actually, the nasopharyngeal swab of her newborn collected upon delivery was positive. Notably, subject no. 17 was the only one to deliver prematurely at week 34. Prematurity was indeed reported to be more frequent in SARS-CoV-2-infected patients[8]. We hypothesize that this might be related to the inflammatory status as a consequence of the viral infection; alternatively, this could have been the result of a pre-existing condition that triggered premature delivery and facili-tated viral entry through the placenta SARS-CoV-2 positivity of umbilical cord plasma from the newborn of subject 17, proving that infection was acquired antenatally by transplacental trans-mission as recently established by the international classification on maternal, foetal and neonatal SARS-CoV-2 infection[32]. Indeed, while the coexistence of the maternal and foetal side in the placenta does not allow us to draw firm conclusions on intrauterine transmission, cord plasma is exclusively foetal material, whose infection may occur solely in utero. In the same woman, the vagina was found to be positive for SARS-CoV-2. Since the presence of the virus in cord blood indicates in utero transmission prior to delivery, we cannot speculate on the risk of acquiring the virus during vaginal delivery in this case. However,

**Table 3 Maternal and foetal SARS-CoV-2 genome and anti-SARS-CoV-2 antibody detection.**

| Subject no. | Clinical outcome | ΔT1−T0 (days) | Maternal plasma Viral RNA | Maternal plasma IgM | Maternal plasma IgG | Vaginal swab Viral RNA | Placenta Viral RNA | Amniotic fluid Viral RNA | Umbilical cord plasma Viral RNA | Umbilical cord plasma IgM | Umbilical cord plasma IgG | Umbilical cord Viral RNA | Nasopharyngeal newborn swab Viral RNA | Milk Viral RNA | Milk IgM | Milk IgG | Classification system by Shah et al.[36] |
|---|---|---|---|---|---|---|---|---|---|---|---|---|---|---|---|---|---|
| 1 | Severe | 2 | − | − | − | − | − | N/A | N/A | N/A | N/A | − | − | + | + | − | Not infected |
| 2 | Mild | 1 | − | − | − | − | − | N/A | − | − | − | − | − | − | + | − | Not infected |
| 3 | Mild | 1 | − | − | − | − | − | − | − | − | − | − | − | − | − | N/A | Not infected |
| 4 | Severe | 2 | + | − | + | − | − | N/A | − | − | − | − | − | N/A | N/A | N/A | Not infected |
| 5 | Mild | 7 | − | − | − | − | − | N/A | − | − | − | − | − | N/A | N/A | N/A | Not infected |
| 6 | Mild | 1 | − | + | + | − | − | N/A | − | − | − | − | − | − | − | − | Not infected |
| 7 | Mild | 12 | − | + | + | − | − | − | − | − | − | − | − | N/A | N/A | N/A | Not infected |
| 8 | Severe | 6 | − | + | + | − | − | N/A | − | − | − | − | − | N/A | N/A | N/A | Not infected |
| 9 | Mild | 1 | N/A | N/A | N/A | − | − | N/A | − | − | − | − | − | − | − | − | Not infected |
| 10 | Mild | 1 | − | − | − | − | − | N/A | − | − | − | − | − | − | − | − | Not infected |
| 11 | Mild | 5 | − | − | − | − | − | N/A | − | − | − | − | − | − | − | − | Not infected |
| 12 | Mild | 4 | − | − | + | − | − | N/A | N/A | − | − | N/A | − | N/A | N/A | N/A | Not infected |
| 13 | Mild | 3 | − | − | + | − | − | N/A | − | − | + | N/A | − | N/A | N/A | N/A | Not infected |
| 14 | Mild | 3 | − | − | + | − | − | N/A | N/A | − | + | N/A | − | N/A | N/A | N/A | Not infected |
| 15 | Mild | 4 | − | − | − | − | − | N/A | N/A | − | − | N/A | − | N/A | N/A | N/A | Not infected |
| 16 | Mild | 2 | − | − | − | − | − | N/A | N/A | − | − | N/A | − | N/A | N/A | N/A | Not infected |
| 17 | Severe | 6 | + | + | + | + | + | N/A | + | − | + | + | + | N/A | N/A | N/A | Confirmed |
| 18 | Mild | 2 | − | − | + | − | − | N/A | N/A | − | + | N/A | − | N/A | N/A | N/A | Not infected |
| 19 | Mild | 9 | − | + | + | − | − | N/A | N/A | − | + | N/A | − | N/A | N/A | N/A | Not infected |
| 20 | Mild | 3 | − | − | + | − | − | N/A | N/A | − | + | N/A | − | N/A | N/A | N/A | Not infected |
| 21 | Mild | 13 | − | − | − | − | + | N/A | N/A | − | − | N/A | − | N/A | N/A | N/A | Not infected |
| 22 | Mild | 10 | − | − | − | − | − | N/A | N/A | − | − | N/A | − | N/A | N/A | N/A | Not infected |
| 23 | Mild | 9 | − | − | − | − | − | − | N/A | − | − | N/A | − | N/A | N/A | N/A | Not infected |
| 24 | Mild | 12 | − | − | + | − | − | N/A | N/A | − | − | N/A | − | N/A | N/A | N/A | Not infected |
| 25 | Mild | 17 | − | + | + | − | + | N/A | N/A | + | + | N/A | + | N/A | N/A | N/A | Possible |
| 26 | Mild | 13 | − | + | + | − | − | N/A | N/A | − | + | N/A | − | N/A | N/A | N/A | Not infected |
| 27 | Mild | 1 | − | − | + | − | − | N/A | N/A | − | + | N/A | − | N/A | N/A | N/A | Not infected |
| 28 | Mild | 3 | − | + | + | − | − | N/A | N/A | − | + | N/A | − | N/A | N/A | N/A | Not infected |
| 29 | Mild | 2 | − | − | + | − | − | N/A | N/A | − | + | N/A | − | − | − | − | Not infected |
| 30 | Mild | 1 | − | − | + | − | − | N/A | N/A | − | + | N/A | − | N/A | N/A | N/A | Not infected |
| 31 | Recovered | N/A | − | − | + | N/A | − | N/A | N/A | − | + | N/A | − | N/A | N/A | N/A | Not infected |

ΔT1−T0 (days) refers to the time spanning between the first COVID19 diagnosis (T0) and delivery (T1).
N/A not available.

**Table 4 Summary of maternal and foetal SARS-CoV-2 genome (a) and anti-SARS-CoV-2 antibody (b) detection.**

| (a) | Maternal plasma (n = 30) | Vaginal swab (n = 31) | Placenta (n = 31) | Umbilical cord plasma (n = 30) | Umbilical cord (n = 12) | Amniotic fluid (n = 6) | Milk (n = 11) |
|---|---|---|---|---|---|---|---|
| Pos, % (n) | 6.7 (2) | 3.2 (1) | 6.4 (2) | 3.3 (1) | 0 (0) | 0 (0) | 9.1 (1) |

| (b) | Maternal plasma (n = 30) | Umbilical cord plasma (n = 30) | Milk (n = 10) |
|---|---|---|---|
| IgM, % (n) | 32.1 (9) | 3.3 (1) | 10 (1) |
| IgG, % (n) | 63.3 (19) | 40.0 (12) | 0 (0) |
| IgM/IgG, % (n) | 32.1 (9) | 3.3 (1) | 0 (0) |

we cannot exclude the possibility of viral intrapartum infection when the virus is present in the vagina. Subject no. 17 delivered 6 days after the first SARS-CoV-2 diagnosis. Most likely due to the short span of $\Delta T1-T0$ time, specific anti-SARS-CoV-2 IgM were not detected in umbilical cord blood. Conversely, subject no. 25, who manifested mild symptoms, was SARS-CoV-2-negative in all the biological samples analysed (maternal plasma, vagina, umbilical cord plasma) but not in the placenta. However, her newborn had a SARS-CoV-2-positive nasopharyngeal swab at birth and both SARS-CoV-2-specific IgM and IgG were detected in umbilical cord plasma. Although still controversial[33], the presence of anti-SARS-CoV-2 IgM strongly suggests SARS-CoV-2 infection in utero[32]. Of note, the positivity of the newborns' nasopharyngeal swabs was not sustained over time, as the following tests were negative. We also detected IgM and IgG in maternal plasma. This is consistent with the span of time between COVID-19 diagnosis and delivery (17 days), where the median of detection of specific IgM/IgG is 13 days[34]. It is important to mention that T0 corresponds to the initial SARS-CoV-2 diagnosis, which does not necessarily coincide with the time of infection. This may result in some cases in SARS-CoV-2-specific antibody detection even though a few days occurred between initial diagnosis and delivery. Therefore, it is not possible to establish an accurate correlation between $\Delta T1-T0$ and the presence of antibodies, either in maternal or foetal blood samples.

We decided to collect all breast milk specimens at 5 days after delivery (T2) for two reasons: the first one related to a greater production of maternal milk, such as not interfering with the feeding of the newborn during the first days of life; the second was related to the higher quantity of antibodies in mature milk. We detected the presence of SARS-CoV-2 RNA in one case only (subject no. 1), which was a severe case. This is consistent with what was previously reported[35]. However, the potential contamination of breast milk by SARS-CoV-2 is still controversial and no univocal consensus has been reached yet[11,36]. Further studies are required to assess whether this represents an infectious and replicative virus or not and whether it may depend on viremia or other factors. It has been previously reported that other β-coronaviruses may pass through milk[37–39]. Although precautions were adopted, we cannot exclude a contamination of the sample by other maternal positive sites. Moreover, we tested milk specimens for the presence of specific anti-SARS-CoV-2 IgM and IgG. We were able to detect IgM in subject no. 1 only. It was previously reported that the absence of IgM and IgG in the milk is not uncommon, especially in the case of respiratory viruses[40]. The protective role of maternal anti-SARS-CoV-2 antibodies has not yet been estimated. This information is urgently required to assess the risk–benefit of breastfeeding and to identify new potential guidelines[36]. A recent study showed the high sensitivity and specificity of the iFlash automated system for antibody detection[41]. However, this methodology has been adapted for the detection of antibodies in milk and the sensitivity may be attenuated on this particular specimen.

Further studies are needed to ascertain long-term outcomes and potential intrauterine vertical transmission in pregnant women infected in the first or second trimester. This observation is even more relevant considering that the temporal and spatial expression of the main SARS-CoV-2 receptor, ACE2, has been reported to change significantly in maternal–foetal interface tissues in the different trimesters[21,42]. We can speculate on the possibility that ACE2 modulation could be directly linked to placenta susceptibility to SARS-CoV-2 infection. Alternatively, we can reason on the possibility that due to altered permeability/ damage of the placenta, probably secondary to an inflammatory status, SARS-CoV-2 is able to bypass the placental barrier and

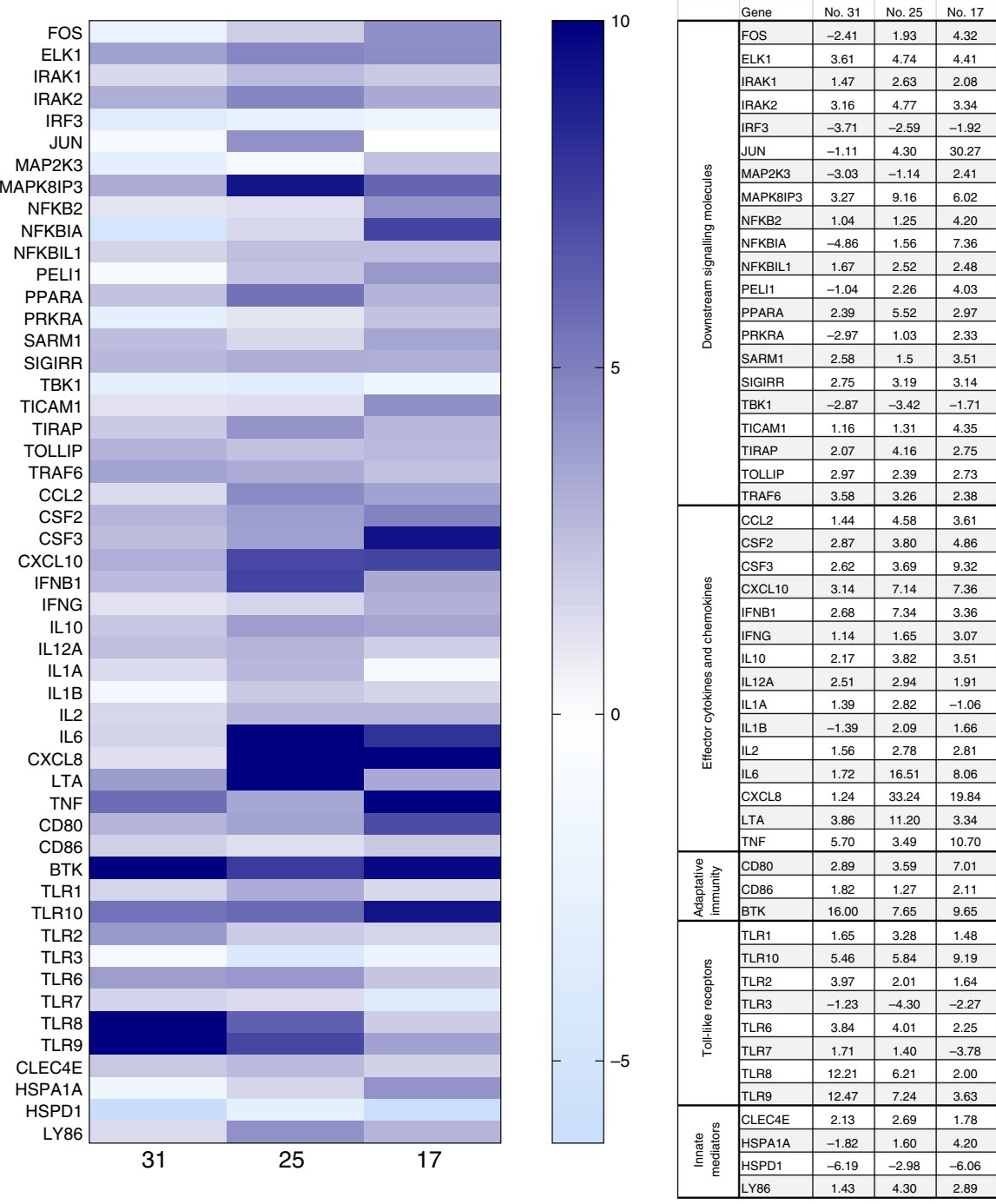

**Fig. 1 mRNA expression of proinflammatory genes is altered in the placentas from SARS-CoV-2-positive subjects.** An 84-gene real-time PCR array was performed on placenta biopsies from subject nos. 31, 25 and 17. The results are shown as a ratio of each SARS-CoV-2-positive subject compared to a SARS-CoV-2-negative subject (CTR−). Gene expression (*n*fold) is shown as a colour scale from light to dark blue. Only targets showing at least >2-fold modulation are shown in the table.

reach foetal blood. This issue still remains to be addressed and further investigated.

As several lines of evidence indicate that systemic maternal infection and consequent inflammation contribute to the disruption of placenta development/function and possibly favour viral vertical transmission[43,44], we decided to profile the inflammatory status of four selected patients at both local (placenta) and systemic (maternal and foetal) levels. The results obtained by different molecular approaches (RNA expression and protein secretion) give us the same take-home message by showing a

trend of generalized immune activation in those patients (17 and 25), who were SARS-CoV-2 positive at delivery and, according to the viral–immunological analyses, infected their neonates in utero. Unexpectedly, this hyperactivation status was far more evident in SARS-CoV-2-negative biological samples (placenta biopsy, maternal and umbilical cord plasma) from subject no. 25, compared to subject no. 17, whose clinical condition was severe. A plausible explanation for this apparent inconsistency stems from the observation that subject no. 17 was undergoing cortisone prophylaxis during the antepartum period that could have

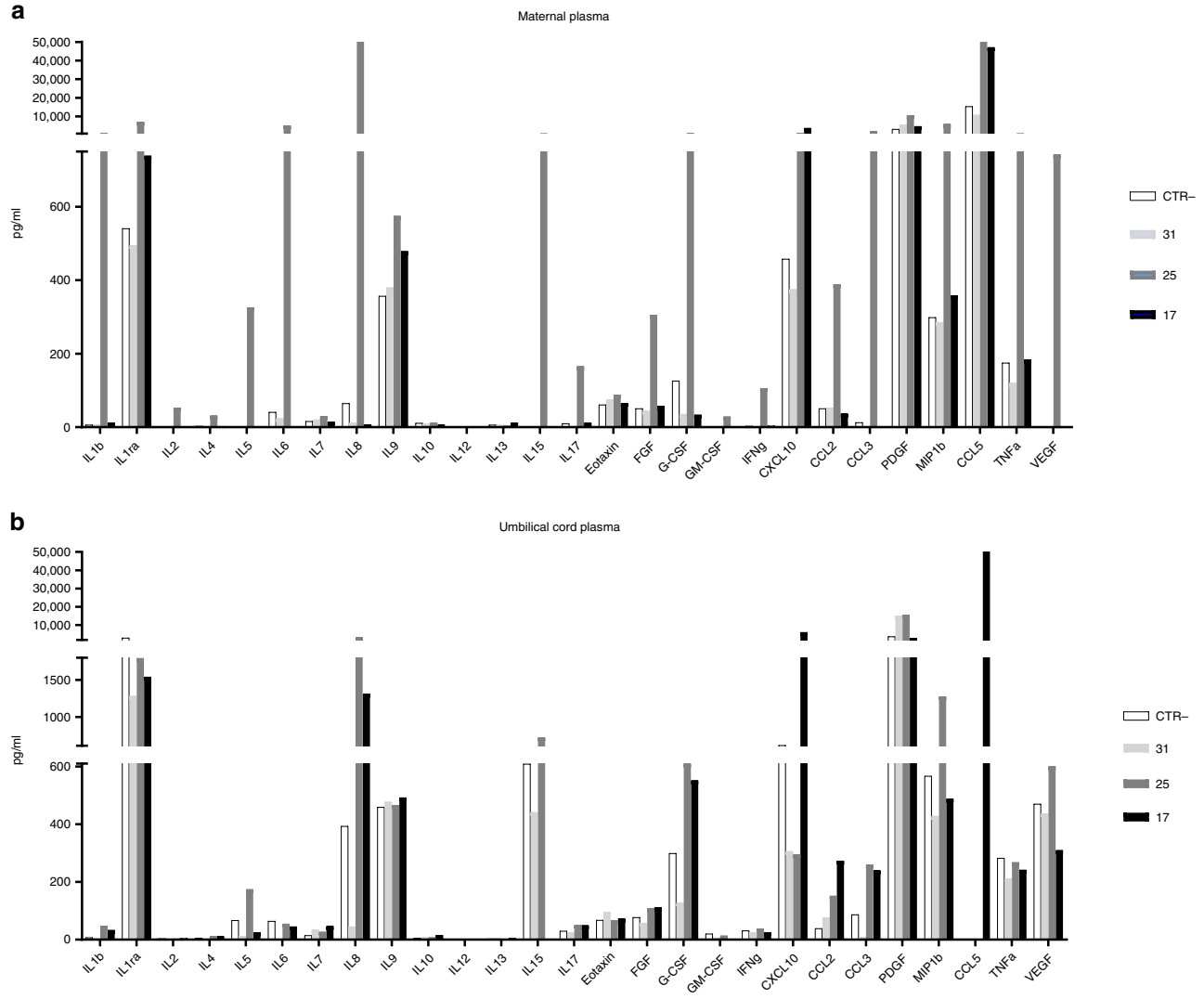

**Fig. 2 Secretion of proinflammatory proteins is altered in the plasma from SARS-CoV-2 -positive subjects.** A 27 cytokine/chemokine multiplex array was performed on **a** maternal and **b** umbilical plasma from subject nos. 31, 25 and 17. As reference, a SARS-CoV-2-negative plasma is shown (CTR−) (n = 4 independent analysed plasma). Protein concentration is shown as pg/ml.

weakened the synthesis and release of inflammatory cytokines/ chemokines. Among the inflammatory factors, their expression was evidently increased in both placenta and cord blood samples from subject nos. 17 and 25; the chemokines CXCL10, CXCL8, CCL5, CCL3 and CCL2 could have played a major role in favouring vertical transmission. Indeed, they could have created a chemotactic gradient between villi and the inter-villous space, where maternal lymphocytes circulate, thus favouring viral dissemination[45]. To perform such molecular analyses, only four subjects (CTR−, nos. 17, 25 and 31) were chosen due to their peculiarities. It is reasonable to presume that such an inflammatory profile may result in multiple placental malfunctions, as recently reported[46]. However, further experiments are envisaged to confirm this distinctive profile and the consequent pathophysiology.

In conclusion, the SARS-CoV-2 genome was detected in umbilical cord plasma, indicating that in utero mother-to-child transmission, although rare, is possible and apparently related to a high maternal and foetal inflammatory state. Although further studies are needed and no firm conclusions can be drawn due to the low number of analysed cases, this should be taken into consideration in the management of COVID-19 pregnant women. Our findings, together with the results obtained by Vivanti et al.[27], deliver an important message that should not be underestimated. The SARS-CoV-2 genome was detected in different biological specimens; nonetheless both Vivanti et al.[27] and our study confirmed that SARS-CoV-2 transplacental infection occurred, according to Shah et al. classification[32]. Indeed, SARS-CoV-2 infection of foetuses, newborns and infants is generally not considered or perceived as having no consequences. The percentage we observed is roughly consistent with the results previously reported by Zeng et al.[47]. Overall, many suspected cases have been reported thus far[11,19,23,26,48–57]. Neonates born to infected mothers must be tested and carefully clinically monitored. Therefore, we encourage the scientific and medical community to deeply consider which guidelines should be more appropriate in clinical practice.

## Methods

**Study population.** This is a prospective multicentre study that includes 31 women: 30 pregnant women with SARS-CoV-2 positive first diagnosis were admitted at delivery (T0) in three COVID-19 maternity hospitals of Lombardy, Italy between March and April 2020. A further woman was admitted to the study (subject no. 31); she was found to be SARS-CoV-2 positive at 32 gestational weeks and delivered at a SARS-CoV-2-free hospital.

All women underwent clinical evaluation of vital signs and symptoms, laboratory analysis and radiological chest assessment at admission at the discretion of physicians. The therapeutic management was consequently tailored according to the clinical findings and national guidelines[58]. Demographic and anthropometric characteristics and medical and obstetric comorbidities were recorded at enrolment through a customized data collection form. All pregnancies were singleton, with a normal course and regular checks, until delivery.

Data on mode of delivery or pregnancy termination, maternal and neonatal outcomes and postpartum clinical evolution (e.g. breastfeeding, skin to skin) were subsequently recorded. Data accuracy was independently verified by two study investigators.

**Specimen collection.** Biological samples were collected at admission (T0), delivery (T1) and postpartum (T2). T0 samples included a nasopharyngeal swab to test positivity for SARS-CoV-2. Nasopharyngeal swabs were obtained following US Centers for Disease Control and Prevention (CDC) guidelines[59]. We inserted the swab into the nostril, parallel to the palate, and then left the swab in place for several seconds to absorb secretions. We then slowly removed the swab while rotating it, as recommended by CDC[59]. At T1, full-thickness placental and umbilical cord biopsies were obtained and a 10-ml venous umbilical cord blood sample was collected in EDTA after the cord was cleaned throughout with a sterile gauze and physiological solution before sampling. Both biopsies and blood samples were obtained in a sterile way by a dedicated operator. In the case of caesarean section, amniotic fluid was collected if possible. We cut the integrity of the lower uterine segment-preserving membranes, and then we withdrew amniotic fluid using a 10-ml sterile syringe; it was not contaminated with blood or meconium.

Moreover, a 10-ml maternal blood sample in EDTA was collected, together with a vaginal swab before labour or caesarean section. Vaginal swabs (obtained following US CDC guidelines[59]) were collected by a dedicated operator who inserted swabs into the vagina up to the vault and rotated the swab in the vaginal vault. This procedure was standardized across participating hospitals. Nasopharyngeal swabs of the babies were collected immediately after vaginal delivery or caesarean section according to the international guidelines[59]. Samples from neonates were collected immediately after vaginal delivery or caesarean section. Neonates were cleaned by dedicated nurses. None of them was in skin-to-skin contact with her/his mother before the nasopharyngeal swab was collected. Neonates were then allowed to room in with their mothers. Five days after delivery (T2), transitional/mature breast milk samples were collected from all breastfeeding women. According to World Health Organization, mothers with SARS-CoV-2 infection can breastfeed their babies using appropriate precautions[60]. Breast milk samples were collected with the same abovementioned safety measures.

For each subject, days occurring between T0 and T1 ($\Delta T1-T0$) were calculated. Samples from obstetrics and gynecology units were immediately transferred to the dedicated laboratory of clinical microbiology, virology and diagnostics of "L. Sacco" Hospital and/or to the laboratory of immunology, University of Milan, according to the kind of specimen, to be readily processed. Alternatively, samples were frozen at −80°C upon collection and transferred to the same laboratories in dry ice.

**Diagnostic analyses.** Molecular analysis was performed to detect viral RNA, using the automated Real-Time PCR ELITe InGenius® system and the GeneFinderTM COVID-19 Plus RealAmp Kit assay (ELITechGroup, France). The reaction mix was prepared according to the manufacturer's instructions. Three target genes, RNA-dependent RNA polymerase (RdRP), nucleocapsid (N) and envelope (E), were simultaneously amplified and tested. A cycle threshold value (Ct-value) <40 was defined as a positive test result according to the manufacturer's instructions.

The presence of SARS-CoV-2-specific antibodies was investigated using SARS-CoV-2 IgG and IgM chemiluminescence immunoassay kits on fully automated iFlash1800 analyser (Shenzen YHLO Biotech Co., Ltd., Shenzen, China): the assay uses nucleocapsid (N) and spike (S) viral proteins as magnetic bead-coating antigens. The value of 10.0 AU/ml was used as the positivity cut-off for IgM, while 7.1 was used for IgG[41]. The limit of detection of this kit is not declared by the company, in accordance with the European Ligand Assay Society. The intra-assay percentage of coefficient variation (%CV) spanned from 2.7 to 5.0 for IgM and the inter-assay %CV spanned from 4.1 to 6.1, while the intra-assay %CV spanned from 2.9 to 4.9 and the inter-assay %CV spanned from 4.0 to 4.9 for IgG.

**Tissue processing.** Placental and umbilical cord biopsies were manually dissected into few sections of approximately 2 mm³. Such sections were then thoroughly homogenized and total RNA was isolated using the acid guanidinium thiocyanate–phenol–chloroform method (RNAbee, Duotech, Milan, Italy). Alternatively, biopsies were paraffin embedded and stored as such.

Plasma samples were collected from the blood of all enrolled subjects, as well as plasma samples from funicular blood, amniotic fluid and vaginal swabs. Moreover, as a control for molecular analyses, plasma from a SARS-CoV-2-negative pregnant woman (CTR−), as well as plasma samples from funicular blood and placental tissues were included. RNA was extracted by the Maxwell® RSC Instrument with the Maxwell® RSC Viral Total Nucleic Acid Purification Kit (Promega, Fitchburg, WI, USA). As a result, RNA eluted in RNase-free water was obtained.

Once RNA was reverse transcribed into cDNA, real-time PCR was performed on a CFX96 (Bio-rad, CA, USA) using TaqMan probes specifically designed to target two regions of the nucleocapsid (N) gene of SARS-CoV-2. For such application, we employed the 2019-nCoV CDC qPCR Probe Assay Emergency Kit (IDT, IA, USA), which also includes primers and probes that target the human RNase P gene (Supplementary Table 1).

**Expression analyses of the inflammatory response.** The inflammatory response was analysed in four selected RNA samples extracted from placenta biopsies of one negative control (CTR−), one SARS-CoV-2 recovered (subject no. 31) subject and two SARS-CoV-2-positive subjects (subject nos. 17 and 25). Subject nos. 17 and 25 gave birth to SARS-CoV-2-positive newborns, according to the first nasopharyngeal swab (T1). Analyses were performed by a PCR array that included a set of 84 optimized real-time PCR primers plus 5 housekeeping genes on a 96-well plates; the procedures suggested by the manufacturer were followed (Qiagen, Hilden, Germany). Undetermined raw CT values were set to 35. Only variables with at least a twofold increase in their value are presented and discussed in the manuscript.

The concentration of 27 cytokines was assessed in maternal and funicular plasma samples from the same four subjects using immunoassays formatted on magnetic beads (Bio-rad, CA, USA) according to the manufacturer's protocol via Luminex 100 technology (Luminex, TX, USA).

**Statistics.** For the study variables, medians and ranges were reported for quantitative variables. All ranges are indicated as minimum–maximum values. Mann–Whitney $U$ test was used as nonparametric test with a $p$ value threshold of 0.05. The analyses were performed using SPSS Statistics, Version 26.0 (IBM Corp. Armonk, NY) together with GraphPad Prism 8.

All the procedures were carried out in accordance with the GLP guidelines adopted in our laboratories as exhaustively detailed in the CDC 2019-Novel Coronavirus (2019-nCoV) Real-Time RT-PCR Diagnostic Panel (https://www.fda.gov/media/134922/download).

**Reporting summary.** Further information on research design is available in the Nature Research Reporting Summary linked to this article.

## Data availability

Source data are provided with this paper. All other data sets generated and analysed in the current study are available from the corresponding author upon reasonable request.

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

## Acknowledgements

We would like to thank Paolo Quaini, Francesco Leone, Federica Fusè, Irma Saulle, Claudia Fusetti, Margherita Longo, Alberto Rizzo, Francesca Romeri, Federica Brunetti, Francesca Sabbatini, Claudia Vanetti and Irma Saulle for their support and contribution to the project. This study was supported by a COVID-19 donation to Obstetrics and Gynecology and to the Laboratory of Immunology, Department of Biomedical and Clinical Sciences, University of Milan, Italy and by a grant from Falk Renewables (REC18GZUCC N. 27767).

## Author contributions

V.S. conceived the presented idea. M.B. further developed the project with the help of C.F. V.S., I.C., P.V., A.S., F.P., C.C. and S.C. performed subject enrolment and clinical management, as well as the sample collection. I.B. was involved in sample collection and management. C.F. and M.B. conceived, planned, performed and analysed the experiments on SARS-CoV-2 genome detection in plasma, biopsies and vaginal swabs and on the inflammatory response. D.T. helped with the interpretation of the data. D.M. and A.M. conceived, planned and performed the experiments on specific antibody detection and experiments on milk under the supervision of M.R.G. C.F., M.B. and V.S. discussed the data and wrote the manuscript. M.C., I.C. and P.V. critically reviewed the manuscript.

## Competing interests

The authors declare no competing interests.

**Ethics declaration**
The protocol was approved by the local Medical Ethical and Institutional Review Board (Milan, area 1, #154082020). We obtained informed written consent from the mothers to perform the procedure and analysis, according to CARE guidelines and in compliance with the Declaration of Helsinki principles.
