## [Peer Review File · Nature Communications]

REVIEWER COMMENTS

Reviewer #1 (Remarks to the Author):

Fenzia et al. report in utero mother-to-fetus transmission of SARS-CoV-2. They analyzed 31 pregnant women and their newborns by viral RNA, immunoglobulin, and inflammatory molecules. Although the nature of the study is important and interesting, more critical evidence is needed to substantiate the conclusions.

Major comments

1. How do authors ensure the positive results reported here are not due to viral RNA and immunoglobulin contamination?
2. Have the authors succeeded in isolating live virus from the specimens?

Minor suggestion

1. In table 3, change "virus" to "viral RNA"

Reviewer #2 (Remarks to the Author):

This is an important report that comes along with the very first demonstration of the transplacental SARS-CoV-2 passage by Vivanti et al in this same journal. Authors should be commended for their work! However there are some points that should be addressed to improve data clarity and value. Further lab work may not be needed but re-interpretation of data, literature update and significant change to the text shall be provided. Please see below:

METHODS

- How were nasopharyngeal swabs performed? We know that a wrong technique may impact on PCR results. Please specify which guidelines and technical details did you follow to perform these swabs.

- Same applies for vaginal swabs. How were these technically performed? Were these vaginal or rectal/vaginal swabs ? ACOG guidelines ? Was this standardised across participating hospitals ?

- Was the cord blood arterial, venous or mixed ? Please specify

- Regarding PCR. I suspect that WHO or ECDC technical guidelines were followed. Please confirm and detail/cite. This would help understanding the reproducibility

- Regarding CLIA. As many of these techniques are available, more details are needed. did you check this technique against others or monoclonal antibodies or just followed manufacturer's instructions. what is the coefficient of variation, the inter-assay and intra-assay accuracy and the limit of detection of this kit ? I see there is a reference in this regard, but it would be better to explicitly detail some of these in the main text for an average reader.

- I suspect that samples from San Matteo University Hospital were only thawed once (when finally analysed). Pls confirm.

- As Placental tissue samples include both the maternal and fetal side, authors should be commended to have taken cord biopsy as well. This is an important confirmation of the viral passage and should be stressed in the text.

- Amniotic fluid samples must be better described. First, were these centrifuged to remove cells and debris before PCR? Second how many were blood or meconium-stained and how were these treated? Were samples taken with intact amniotic membranes or not ? this is important for the classification of your cases (see below). Pls add at least some of these details

- It would have been nice to couple the inflammatory analysis of placenta and cord biopsies with at least basic histology or immunohistochemistry. this would have provided visual report of what you measured, also because you have both positive and control samples in your series. However this has been already done by Vivanti et al and it may be unnecessary to add. I would leave this to the Editor to decide.

STATISTICS

- We need more details. I guess that 'range' refers to interquartile range (25-75th percentile) but pls specify. Nonparametric methods seem to have been used: please specify the test and the significance threshold. Pls add more details regarding time effect analysis (see below).

RESULTS

- We absolutely need to classify babies infected from SARS-CoV-2 according to the international classification on maternal, fetal and neonatal SARS-CoV-2 infection. See here : Shah PS et al. Classification system and case definition for SARS-CoV-2

infection in pregnant women, fetuses, and neonates. Acta Obstet Gynecol Scand 2020 - DOI: 10.1111/aogs.13870

It seems that there are only 2 infected babies (first NP swab positive). However please look if there are others according to IgM and other PCR on cord, placenta and amniotic fluid that you did. All these data contributed to the aforementioned classification and can identify other infected babies.

The results of these classification must be reported in results, either in the main text or as a pie graph if there are many cases.

By the way, in table 2 you cannot write 'newborn COVID' , unless these infants were really symptomatic. Pls only write " infected neonates".

- Consistently, pls add clinical data of all neonates that turned out to be infected according to the classification above. Some of them may present clinical symptoms. In this case they become cases of neonatal COVID. there are actually many described in the literature with a wide spectrum of clinical features, although usually with mild evolution. It is important to describe this or just say that there are no clinical manifestation.

If there are any symptoms or signs , the way to report them must be as open as possible (ie: do not exclude any just because they seem related to other neonatal disorders).

-In results: do not say "COVID infection" but " SARS-CoV-2 infection". COVID is a disease and can only be evident if there are signs and/or symptoms .. a swab is not enough.

- By reading further it seems that only there are only the two cases of infected neonates. Since they have at least placental PCR positive, they are likely to be classified as vertical intra-uterus transmission. Pls again look at the classification above and specify.

The same has been done by Vivanti et al in the first paper published in Nat Commun 2020 (reference n.25 in your paper) demonstrating the transplacental transmission of SARS-CoV-2 and, according to

that classification, their case was classified as fully congenital. You should compare the two results and comment on this.

DISCUSSION

- The study is clearly important but cannot answer some important questions. For instance, we cannot say almost nothing about human milk transmission. In fact only few milk samples have been analysed and not in a serial way. Only one turned out positive after two days (so a short time frame) and one may wonder if there is a sort of viral clearance afterwards. Moreover that particular case did not present maternal viremia. This is again a clear confirmation that there are still unknown issues in SARS-CoV-2 biology since the passage into a particular tissue or biological fluid may depend not only on viremia but on other factors (receptors density, local inflammation, genetic predisposition, others...).

Some of these concepts must be commented in discussion and in study limitation also recalling that there have been conflicting data on the viral presence in human milk (just to name two: Groß, R. et al. Detection of SARS-CoV-2 in human breastmilk. *Lancet* 395, 1757–1758 (2020) and Chen, H. et al. Clinical characteristics and intrauterine vertical transmission potential of COVID-19 infection in nine pregnant women: a retrospective review of medical records. *Lancet* 395, 809–815 (2020)).

Also recall that other Beta-coronavirus may pass through animal milk (MERS: van Doremalen N, Bushmaker T, Karesh WB, Munster VJ. Stability of Middle East respiratory syndrome coronavirus in milk. *Emerg Infect Dis.* 2014 ;20(7):1263-4. doi: 10.3201/eid2007.140500. - Porcine Epidemic Diarrhea Virus and Bovine Coronavirus: Langel SN, Wang Q, Vlasova AN, Saif LJ. Host Factors Affecting Generation of Immunity Against Porcine Epidemic Diarrhea Virus in Pregnant and Lactating Swine and Passive Protection of Neonates. *Pathogens.* 2020;9(2). pii: E130. doi: 10.3390/pathogens9020130 and Decaro N, Mari V, Desario C, et al. Severe outbreak of bovine coronavirus infection in dairy cattle during the warmer season. *Vet Microbiol.* 2008;126(1-3):30-9).

- Please modify your discussion. Your study although interesting is NOT the first one reporting SARS-CoV-2 in placentas nor in cord blood. The following ones have already did so even adding other techniques (histology, immunohistochemistry, TEM):

Vivanti AJ., Vauloup-Fellous C., Prevot S., Zupan V., Suffee C., Do Cao J., Benachi A., De Luca D. Transplacental transmission of SARS-CoV-2 infection. *Nat Commun.* 2020 Jul 14;11(1):3572. doi: 10.1038/s41467-020-17436-6

Baud, D. et al. Second-Trimester Miscarriage in a Pregnant Woman With SARS-CoV-2 Infection. *JAMA* 323, 2198 (2020).

Algarroba, G. N. et al. Visualization of SARS-CoV-2 virus invading the human placenta using electron microscopy. *Am. J. Obstet. Gynecol.* (2020) doi:10.1016/j.ajog.2020.05.023.

Hosier, H. et al. SARS-CoV-2 Infection of the Placenta. *medRxiv* 2020.04.30.20083907 (2020) doi:10.1101/2020.04.30.20083907.

Patanè, L. et al. Vertical transmission of coronavirus disease 2019: severe acute respiratory syndrome coronavirus 2 RNA on the fetal side of the placenta in pregnancies with coronavirus disease 2019–positive mothers and neonates at birth. *Am. J. Obstet. Gynecol. MFM* 100145 (2020) doi:10.1016/j.ajogmf.2020.100145.

Sisman, J. et al. Intrauterine transmission of SARS-CoV-2 infection in a preterm infant *Pediatr. Infect. Dis. J. Online First*, (2020).

- I wonder if it is possible to measure sIgA in human milk. To the best of my knowledge no specific kits have been released to this end. Pls elaborate on this.

- You reported 2/31 positive babies. If this is confirmed by the classification (see above), it will give a rough vertical infection rate of 10%. this is not so low and is consistent with earlier Chinese data (Zeng, L. et al. Neonatal Early-Onset Infection With SARS-CoV-2 in 33 Neonates Born to Mothers With COVID-19 in Wuhan, China. *JAMA Pediatr.* (2020) doi:10.1001/jamapediatrics.2020.0878).

I believe that the main (and very important message) to give here is that vertical transmission DOES exist. It is quite rare and, fortunately, neonatal COVID is milder than the disease in adults but this is a real problem. Unfortunately, the message that is still being diffused is that the virus does not affect babies and this is not totally true. We should have said from the beginning " we do not know". With the first multi-technique demonstration by Vivanti et al (see above) and your data, on top of more than 100 neonatal suspected cases in the literature, there is enough to recommend a change in clinical guidelines. Neonates born to infected mothers must be tested and carefully clinically monitored. You must stress all this in conjunction with the Vivanti's data.

The rest of discussion may be shortened, especially in the parts where hypothesis and possible speculation are given (not so useful when we talk about just two cases).

- Table 1. pls specify what type of antiviral has been given. If this was remdesivir, it can impact of viral replication and on your results. This should be acknowledge in discussion, study limitations.

- Line 216.. not placentae but placentas

Reviewer #3 (Remarks to the Author):

Introduction:

1. This will need to be modified significantly as it appears that authors have not done a careful review of the literature. Several case reports of probable and possible mother-child transmission has been reported.

a. PMID: 32409520

b. PMID: 32580461

c. PMID: 32425663

d. PMID: 32332320

e. PMID: 32704477

f. AND PROBABLY MANY MORE.....

Methods:

1. A detailed description of when and how sample from baby was collected immediately after birth. Any skin to skin contact with mother? How were the babies kept – separated/with mothers? Were babies washed/cleaned before sampling?

2. Detailed description of breastmilk samples – what was the hygienic measures employed before collecting sample? Were all samples collected at same time or any time in 10 days – this decreases likelihood of positivity if not collected at closer time to birth as the viral load in maternal circulation will go down anyway.

Results:

1. Tables 3 and 4 – is table 4 a summary of results of table 3 – then no need to repeat and could be described easily in text.

2. Change all indication of “viral” detection to “detection of viral genome” as the test does not dictate virus.

3. There is no surprising finding that viral genome was detected in vaginal swab – it has been described – PMID 32409520

4. Placental injury has also been described in form of maternal vascular malperfusion chronic intervillitis associated with COVID-19 infection. (PMID: 32692408 and several others). How do authors speculate their results are indicative or differentiable from MVM and angiosis versus inflammation?

Discussion:

1. In light of all the references mentioned above, the discussion will need to be modified to remove all assertions of “first” for anything.

2. The neonates who are identified to be positive will need to be classified using a proper system of classification with more details on their testing as without which it is difficult to say it is congenital/intrapartum or postpartum transmission. PMID: 32277845

We would like to thank the reviewers for the overall positive assessments of our manuscript and for the detailed critical comments that we believe led to improvements in our manuscript and therefore enhanced its potential interest. Moreover, we would like to thank the reviewers for the promptness of the revision, which we believe is critical, given the current times. Please, find below all the revisions and changes we produced into the manuscript entitled "IN- UTERO MOTHER-TO-CHILD SARS-CoV-2 TRANSMISSION: viral detection and fetal immune response" NCOMMS-20-28958, as rebuttal letter.

REVIEWER COMMENTS

Reviewer #1 (Remarks to the Author):

Fenzia et al. report in utero mother-to-fetus transmission of SARS-CoV-2. They analyzed 31 pregnant women and their newborns by viral RNA, immunoglobulin, and inflammatory molecules. Although the nature of the study is important and interesting, more critical evidence is needed to substantiate the conclusions.

Major comments

1. How do authors ensure the positive results reported here are not due to viral RNA and immunoglobulin contamination?

All the biological samples analysed for SARS-CoV-2 genome presence were collected with precautions to avoid pre-analytical contamination as described in the sampling collection paragraph in the method section.

Furthermore, in order to ensure neither positive nor negative false PCR results the operators adopted all the good laboratory practice in compliance with the biosafety standards which are exhaustively detailed in the CDC 2019-Novel Coronavirus (2019-nCoV) Real-Time RT-PCR Diagnostic Panel (<https://www.fda.gov/media/134922/download>). To summarize:

- separate areas for assay setup and handling of nucleic acids were maintained.
- aerosol barrier pipette tips were change between all manual liquid transfers.
- separate, dedicated equipment (e.g., pipettes, microcentrifuges) and supplies (e.g., microcentrifuge tubes, pipette tips) were maintained for assay setup and handling of extracted nucleic acids.
- clean lab coat and powder-free disposable gloves were worn when setting up assays.
- gloves were changed between samples and whenever contamination was suspected.
- Work surfaces, pipettes, and centrifuges were cleaned and decontaminated with cleaning products to minimize risk of nucleic acid contamination.
- RNA was maintained on a cold block or on ice during preparation and use to ensure stability.
- every analysis session has been performed using the negative control which showed negative/no amplification signal.

Moreover, as for the primer assay employed in this study both N1 and N2 forward and reverse primers showed no sequence homology with SARS coronavirus and combining primers and probe, no significant homologies with human genome, other coronaviruses or human microflora that would predict potential false positive rRT-PCR results were observed.

As for immunoglobulin contamination, a pre-analytical contamination of milk could have been occurred if we consider a concomitant ulceration or bleeding through the nipple. However, we have excluded this possibility considering the significant divergency in IgM titer detected in milk and in mother's serum (12,24 vs 1,7AU/ml).

For additional transparency, the sentence "as exhaustively detailed in the CDC 2019-Novel Coronavirus (2019-nCoV) Real-Time RT-PCR Diagnostic Panel (<https://www.fda.gov/media/134922/download>)." was added at the end of the method section.

2. Have the authors succeeded in isolating live virus from the specimens?

Thank you for the stimulating input. The aim of the authors was just to verify whether the viral genome was present or not in the biological samples analysed. Therefore, the authors did not isolate the virus from any of the biological samples which tested positive to SARS-CoV-2 PCR. However, failure in viral isolation is not indicative of viral absence nor its inability to replicate. SARS-CoV-2 genome detection in multiple biological samples (Plasma, Vagina, Placenta, umbilical cord plasma etc.) from the same patient/newborn in the two described SARS-CoV-2 positive cases suggests that the virus is able to replicate and spread in different anatomical districts as well as to pass through the placenta and infect the foetus antepartum. Moreover, according to the recently published international classification on maternal, fetal and neonatal SARS-CoV-2 infection (Shah PS et al. Classification system and case definition for SARS-CoV-2 infection in pregnant women, fetuses, and neonates. *Acta Obstet Gynecol Scand* 2020 - DOI: 10.1111/aogs.13870), the detection of the virus by PCR from cord blood is sufficient to confirm infection.

Minor suggestion

1. In table 3, change "virus" to "viral RNA"

According to the reviewer suggestion, in table 3 virus was changed into "Viral RNA".

Reviewer #2 (Remarks to the Author):

This is an important report that comes along with the very first demonstration of the transplacental SARS-CoV-2 passage by Vivanti et al in this same journal. Authors should be commended for their work! However there are some points that should be addressed to improve data clarity and value. Further lab work may not be needed but re-interpretation of data, literature update and significant change to the text shall be provided. Please see below:

METHODS

- How were nasopharyngeal swabs performed? We know that a wrong technique may impact on PCR results. Please specify which guidelines and technical details did you follow to perform these swabs. Thank you for your question. We added the phrase in the method section: "Nasopharyngeal swabs were obtained following US Center for Disease Control and Prevention guidelines. We inserted the swab into the nostril, parallel to the palate, then we left the swab in place for several seconds to absorb secretions and then slowly removed the swab while rotating it, as recommended by CDC."

Moreover, we added the following reference: Centers for Disease Control and Prevention. Interim guidelines for collecting, handling, and testing clinical specimens from persons for coronavirus disease 2019 (COVID-19). April 14, 2020 (<https://www.cdc.gov/coronavirus/2019-ncov/lab/guidelines-clinical-specimens.html>).

- Same applies for vaginal swabs. How were these technically performed? Were these vaginal or rectal/vaginal swabs ? ACOG guidelines ? Was this standardised across participating hospitals ?

We added the phrase in the method section: “Vaginal swabs (obtained following US Center for Disease Control and Prevention guidelines) were collected before labor or caesarean section by a dedicated operator who inserted swab into the vagina up to the vault and rotated the swab in the vaginal vault. This procedure was standardised across participating hospitals.”

- Was the cord blood arterial, venous or mixed ? Please specify

We add the phrase in the methods section: “We collected 10 mL of venous umbilical cord blood in EDTA.”

- Regarding PCR. I suspect that WHO or ECDC technical guidelines were followed. Please confirm and detail/cite. This would help understanding the reproducibility

The reviewer is correct. All PCR procedure were performed according to the ECDC technical guidelines, as consultable at <https://www.fda.gov/media/134922/download> and cited in the manuscript.

- Regarding CLIA. As many of these techniques are available, more details are needed. did you check this technique against others or monoclonal antibodies or just followed manufacturer's instructions. what is the coefficient of variation, the inter-assay and intra-assay accuracy and the limit of detection of this kit ? I see there is a reference in this regard, but it would be better to explicitly detail some of these in the main text for an average reader.

We followed manufacturer's instructions. For further clarification, the following sentence was added to the manuscript in the method section: “Limit of detection of this kit are not declared by the company, in accordance with the European Ligand Assay Society. The intra-assay %CV is spanning from 2.7 to 5.0 for IgM and inter-assay %CV is spanning from 4.1 to 6.1, while intra-assay %CV is spanning from 2.9 to 4.9 and inter-assay %CV is spanning from 4.0 to 4.9 for IgG.”

- I suspect that samples from San Matteo University Hospital were only thawed once (when finally analysed). Pls confirm.

We confirm that samples from S. Gerardo Hospital/MBBM Foundation (Monza), and San Matteo (Pavia) were thawed just once to be analysed. In that occasion three independent sampling were performed for each placenta biopsy and they were stored in RNA Later in order to preserve the material for further analyses. However, following Nature Communication policies, we were asked to remove any sensitive information, including hospital's names.

- As Placental tissue samples include both the maternal and fetal side, authors should be commended to have taken cord biopsy as well. This is an important confirmation of the viral passage and should be stressed in the text.

We thank the Reviewer for her/his positive comment on the sampling plan adopted in this study. Indeed, the choice to collect umbilical cord biopsies and blood was related to the necessity to analyse a tissue of exclusive fetal origin, an essential condition to ascertain in-utero SARS-CoV-2 transmission. As suggested by the reviewer the relevance of this result has been further stressed in the discussion section: “SARS-CoV-2 positivity of umbilical cord plasma from subject's 17 newborn proves that infection was acquired antenatally by transplacental transmission as recently established by the international classification on maternal, fetal and neonatal SARS-CoV-2 infection (DOI: 10.1111/aogs.13870). Indeed, while the co-existence of the maternal and fetal side in placenta does not allow to draw firm conclusions on intra-uterine transmission, cord plasma is exclusively fetal material whose infection may occur solely in-utero.”

- Amniotic fluid samples must be better described. First, were these centrifuged to remove cells and debris before PCR? Second how many were blood or meconium-stained and how were these treated?

Were samples taken with intact amniotic membranes or not? this is important for the classification of your cases (see below). Pls add at least some of these details

We added the phrase in the methods section: “We collected amniotic fluid only in case of caesarean section. We cut the lower uterine segment preserving membranes’ integrity and then we withdraw amniotic fluid using a 10-mL sterile syringe, it was not contaminated with blood or meconium.”

- It would have been nice to couple the inflammatory analysis of placenta and cord biopsies with at least basic histology or immunohistochemistry. this would have provided visual report of what you measured, also because you have both positive and control samples in your series. However this has been already done by Vivanti et al and it may be unnecessary to add. I would leave this to the Editor to decide.

I agree with you that adding placental histopathological information over inflammatory data could strengthen our study. Nevertheless, adding these few histopathological data would not add a significant piece of information. Indeed, Vivanti et al. already published such result. Nevertheless, I am glad to inform you that we are already working on a wide case-control study on placental COVID19-related histopathology. Together with our unit of pathology, we are enrolling several SARS-CoV-2 positive pregnant patients, to study placental characteristics, vs SARS-CoV-2 negative pregnant women.

STATISTICS

- We need more details. I guess that 'range' refers to interquartile range (25-75th percentile) but pls specify. Nonparametric methods seem to have been used: please specify the test and the significance threshold. Pls add more details regarding time effect analysis (see below).

We added the following sentence in the method section: “All ranges are indicated as minimum-maximum values. Mann Whitney u test was used as non-parametric test with a P value threshold of 0.05”.

Concerning the time effect analyses, we would like to point out that the T0 corresponds to the initial SARS-CoV-2 diagnosis, which does not necessarily coincide with the time of infection. This may result in some cases in SARS-CoV-2-specific antibody detection even though few days occurred between initial diagnosis and deliver. Therefore, it is not possible to establish an accurate correlation between $\Delta T1-T0$ and the presence of antibodies, either in maternal or fetal blood samples. However, we believe that this is an interesting piece of information, as it lets us speculate about the presence of IgM in the umbilical cord blood in patient n.25, whose $\Delta T1-T0$ was 17 days, as mentioned in the discussion section.

Following the Reviewer #2 suggestion, more details were added in the manuscript. In particular: in the study population paragraph of the method section “30 pregnant women with SARS-CoV-2 positive first diagnosis were admitted at delivery (T0) in three COVID-19 maternity hospitals” and in the discussion section “It is important to mention that T0 corresponds to the initial SARS-CoV-2 diagnosis, which does not necessarily coincide with the time of infection. This may result in some cases in SARS-CoV-2-specific antibody detection even though few days occurred between initial diagnosis and deliver. Therefore, it is not possible to establish an accurate correlation between $\Delta T1-T0$ and the presence of antibodies, either in maternal or fetal blood samples.”

RESULTS

- We absolutely need to classify babies infected from SARS-CoV-2 according to the international classification on maternal, fetal and neonatal SARS-CoV-2 infection. See here : Shah PS et al. Classification system and case definition for SARS-CoV-2 infection in pregnant women, fetuses, and neonates. Acta Obstet Gynecol Scand 2020 - DOI: 10.1111/aogs.13870

Thank you for your suggestion. According to the International classification on maternal, fetal and neonatal SARS-CoV-2 infection we can classify babies infected from SARS-CoV-2 as follows:

“Subject n.17 newborn could be classified as a confirmed congenital infection because of the detection of the virus by PCR in a nasopharyngeal swab at birth (collected after cleaning baby), in placental sample and in umbilical cord plasma. Subject n.25 could be classified as a possible neonatal infection acquired intrapartum because of the detection of the virus by PCR in nasopharyngeal swab at birth (collected after cleaning baby), but not at 24-48 hours of age, and SARS-CoV-2 specific antibodies in the umbilical cord plasma.”

We added the suggested neonate classification in results section and in bibliography (Shah PS et al.) and in Table n. 3.

It seems that there are only 2 infected babies (first NP swab positive). However please look if there are others according to IgM and other PCR on cord, placenta and amniotic fluid that you did. All these data contributed to the aforementioned classification and can identify other infected babies. According to the international classification on maternal, fetal and neonatal SARS-CoV-2 infection the results obtained in this study suggest that infection is confirmed in one baby (detection of the virus by PCR in umbilical cord blood) and probable in another one (detection of the virus by PCR in nasopharyngeal swab at birth, collected after cleaning baby, placental swab and presence of anti-SARS-CoV-2 IgM antibodies in umbilical cord blood). All the other new-borns were not infected according to these guidelines

The results of these classification must be reported in results, either in the main text or as a pie graph if there are many cases.

To endorse the international classification guidelines, we partially modified Table 3 by adding the results relative to viral detection in amniotic fluid and new-borns' nasopharyngeal swab plus a further column reporting the category of infection likelihood assigned to each neonate.

Furthermore, the results revised according to the criteria of international classification have been commented in the discussion section.

By the way, in table 2 you cannot write 'newborn COVID' , unless these infants were really symptomatic. Pls only write "infected neonates".

According to the reviewer suggestion “newborn COVID” was changed with “infected neonates”

- Consistently, pls add clinical data of all neonates that turned out to be infected according to the classification above. Some of them may present clinical symptoms. In this case they become cases of neonatal COVID. there are actually many described in the literature with a wide spectrum of clinical features, although usually with mild evolution. It is important to describe this or just say that there are no clinical manifestation.

If there are any symptoms or signs, the way to report them must be as open as possible (ie: do not exclude any just because they seem related to other neonatal disorders).

The sentence “All the neonates were healthy and the two SARS-CoV-2 positive babies were totally asymptomatic” is now added in the result section.

-In results: do not say "COVID infection" but " SARS-CoV-2 infection". COVID is a disease and can only be evident if there are signs and/or symptoms. a swab is not enough.

We do apologize for the oversight. The reviewer is correct and according to her/his caveat “COVID infection” was replaced by “SARS-CoV-2 infection” all along the manuscript.

- By reading further it seems that only there are only the two cases of infected neonates. Since they have at least placental PCR positive, they are likely to be classified as vertical intra-uterus transmission. Pls again look at the classification above and specify.

The same has been done by Vivanti et al in the first paper published in Nat Commun 2020 (reference n.25 in your paper) demonstrating the transplacental transmission of SARS-CoV-2 and, according to

that classification, their case was classified as fully congenital. You should compare the two results and comment on this.

We considered the international classification as suggested by the reviewer and included the results in the manuscript, as above-mentioned. Moreover, in the discussion section, we compared our results with those obtained by Vivanti et al.: “Our findings, together with the results obtained by Vivanti et al.³², deliver an important message that should not be underestimated. SARS-CoV-2 genome was detected in different biological specimens, nonetheless both Vivanti et al. and our study confirm that SARS-CoV-2 transplacental infection occurred, according to Shah et al. classification”

DISCUSSION

- The study is clearly important but cannot answer some important questions. For instance, we cannot say almost nothing about human milk transmission. In fact only few milk samples have been analysed and not in a serial way. Only one turned out positive after two days (so a short time frame) and one may wonder if there is a sort of viral clearance afterwards. Moreover that particular case did not present maternal viremia. This is again a clear confirmation that there are still unknown issues in SARS-CoV-2 biology since the passage into a particular tissue or biological fluid may depend not only on viremia but on other factors (receptors density, local inflammation, genetic predisposition, others...).

Some of these concepts must be commented in discussion and in study limitation also recalling that there have been conflicting data on the viral presence in human milk (just to name two: Groß, R. et al. Detection of SARS-CoV-2 in human breastmilk. *Lancet* 395, 1757–1758 (2020) and Chen, H. et al. Clinical characteristics and intrauterine vertical transmission potential of COVID-19 infection in nine pregnant women: a retrospective review of medical records. *Lancet* 395, 809–815 (2020)). Also recall that other Beta-coronavirus may pass through animal milk (MERS: van Doremalen N, Bushmaker T, Karesh WB, Munster VJ. Stability of Middle East respiratory syndrome coronavirus in milk. *Emerg Infect Dis.* 2014 ;20(7):1263-4. doi: 10.3201/eid2007.140500. - Porcine Epidemic Diarrhea Virus and Bovine Coronavirus: Langel SN, Wang Q, Vlasova AN, Saif LJ. Host Factors Affecting Generation of Immunity Against Porcine Epidemic Diarrhea Virus in Pregnant and Lactating Swine and Passive Protection of Neonates. *Pathogens.* 2020;9(2). pii: E130. doi: 10.3390/pathogens9020130 and Decaro N, Mari V, Desario C, et al. Severe outbreak of bovine coronavirus infection in dairy cattle during the warmer season. *Vet Microbiol.* 2008;126(1-3):30-9).

We would like to thank Reviewer #2 for her/his effort to improve our manuscript and making the discussion more comprehensive and stimulating. The discussion section was implemented as follows: “This is consistent with what previously reported. However, the potential contamination of breastmilk by SARS-CoV-2 is still controversial and no univocal consensus has been reached yet. Further studies are required to assess whether this represents an infectious and replicative virus or not, and whether it may depend on viremia or other factors. It has been previously reported that other β -coronaviruses may pass through milk. Although precautions were adopted, we cannot exclude a contamination of the sample by other maternal positive sites. [...] The protective role of maternal anti-SARS-CoV-2 antibodies has not been estimated yet. This information is urgently required in order to assess the risk-benefit of breastfeeding and identify new potential guidelines” and all the suggested citations and more were included and briefly discussed.

- Please modify your discussion. Your study although interesting is NOT the first one reporting SARS-CoV-2 in placentas nor in cord blood. The following ones have already did so even adding other techniques (histology, immunohistochemistry, TEM):

Vivanti AJ., Vauloup-Fellous C., Prevot S., Zupan V., Suffee C., Do Cao J., Benachi A., De Luca D. Transplacental transmission of SARS-CoV-2 infection. *Nat Commun.* 2020 Jul 14;11(1):3572. doi:

10.1038/s41467-020-17436-6

Baud, D. et al. Second-Trimester Miscarriage in a Pregnant Woman With SARS-CoV-2 Infection. *JAMA* 323, 2198 (2020).

Algarroba, G. N. et al. Visualization of SARS-CoV-2 virus invading the human placenta using electron microscopy. *Am. J. Obstet. Gynecol.* (2020) doi:10.1016/j.ajog.2020.05.023.

Hosier, H. et al. SARS-CoV-2 Infection of the Placenta. *medRxiv* 2020.04.30.20083907 (2020) doi:10.1101/2020.04.30.20083907.

Patanè, L. et al. Vertical transmission of coronavirus disease 2019: severe acute respiratory syndrome coronavirus 2 RNA on the fetal side of the placenta in pregnancies with coronavirus disease 2019–positive mothers and neonates at birth. *Am. J. Obstet. Gynecol. MFM* 100145 (2020) doi:10.1016/j.ajogmf.2020.100145.

Sisman, J. et al. Intrauterine transmission of SARS-CoV-2 infection in a preterm infant *Pediatr. Infect. Dis. J. Online First*, (2020).

Thank you for your suggestion. All the above mentioned papers and some more have been added. However, we did not find any of these reporting a positive umbilical cord blood. In fact, Hosier et al. are the only one reporting the presence of SARS-CoV-2 in a homogenized umbilical cord tissue, but not in the umbilical cord blood. In our opinion this makes quite a difference, as the blood itself may spread the virus to other fetal tissues. Therefore, the opening statement of the discussion has been changed as follows: “We report for the first time that SARS-CoV-2 is found in the umbilical cord blood. Also, consistently with the literature, we found the virus in the vagina of a pregnant woman and in at-term placentas.”

- I wonder if it is possible to measure sIgA in human milk. To the best of my knowledge no specific kits have been released to this end. Pls elaborate on this.

To our knowledge, no specific kit to measure sIgA in human milk are available so far, but viral ELISA assays developed and validated for plasma may be adapted to test human milk. Indeed, a recent report found a strong sIgA antibody SARS-CoV-2 immune response in breastmilk from 12 out of 15 mothers (80%) previously infected with Covid-19 (ox A, Marino J, Amanat F, Krammer F, Hahn-Holbrook J, Zolla-Pazner S, et al. Evidence of a significant secretory-IgA dominant SARS-CoV-2 immune response in human milk following recovery from COVID-19. *medRxiv* 2020; <https://doi.org/10.1101/2020.05.04.200899959>).

- You reported 2/31 positive babies. If this is confirmed by the classification (see above), it will give a rough vertical infection rate of 10%. this is not so low and is consistent with earlier Chinese data (Zeng, L. et al. Neonatal Early-Onset Infection With SARS-CoV-2 in 33 Neonates Born to Mothers With COVID-19 in Wuhan, China. *JAMA Pediatr.* (2020) doi:10.1001/jamapediatrics.2020.0878). I believe that the main (and very important message) to give here is that vertical transmission DOES exist. It is quite rare and, fortunately, neonatal COVID is milder than the disease in adults but this is a real problem. Unfortunately, the message that is still being diffused is that the virus does not affect babies and this is not totally true. We should have said from the beginning " we do not know". With the first multi-technique demonstration by Vivanti et al (see above) and your data, on top of more than 100 neonatal suspected cases in the literature, there is enough to recommend a change in clinical guidelines. Neonates born to infected mothers must be tested and carefully clinically monitored. You must stress all this in conjunction with the Vivanti's data.

The rest of discussion may be shortened, especially in the parts were hypothesis and possible speculation are given (not so useful when we talk about just two cases).

We are pleased to realize that our work is well accepted and considered solid and influential. We are thankful to the Reviewer #2 for her/his forthcoming comments and the incentive to deliver an important take-home message. Consequently, we modified the last section of the discussion as follows: “In conclusion, for the first time SARS-CoV-2 genome was detected in umbilical cord plasma, indicating that in-utero mother-to-child transmission, although rare, is possible and apparently related to a high maternal and fetal inflammatory state. Although further studies are needed and no firm conclusion can be drawn due to the low number of analysed cases, this should be taken into consideration in the management of COVID-19 pregnant women. Our findings, together with the results obtained by Vivanti et al., deliver an important message that should not be underestimated. SARS-CoV-2 genome was detected in different biological specimens, nonetheless both Vivanti et al. and our study confirm that SARS-CoV-2 transplacental infection occurred, according to Shah et al. classification. Indeed, SARS-CoV-2 infection of fetuses, newborns and infants is generally not considered or perceived as with no consequences. The percentage we observed is roughly consistent with the results previously reported by Zeng et al. and overall many suspected case have been reported so far. Neonates born to infected mothers must be tested and carefully clinically monitored. Therefore, we encourage scientific and medical community to deeply consider which guidelines should be more appropriate in the clinical practice”

- Table 1. pls specify what type of antiviral has been given. If this was remdesivir, it can impact of viral replication and on your results. This should be acknowledge in discussion, study limitations. We administered only Lopinavir/Ritonavir as antiviral drugs, no woman was given Remdesivir. Table 1 was modified accordingly.

- Line 216.. not placentae but placentas
The word placentae was corrected throughout the manuscript

Reviewer #3 (Remarks to the Author):

Introduction:

1. This will need to be modified significantly as it appears that authors have not done a careful review of the literature. Several case reports of probable and possible mother-child transmission has been reported.

- a. PMID: 32409520
- b. PMID: 32580461
- c. PMID: 32425663
- d. PMID: 32332320
- e. PMID: 32704477
- f. AND PROBABLY MANY MORE.....

We would like to thank the Reviewer #3 for her/his effort in keeping our manuscript up-to-date. As the field is quickly evolving, especially during this last month, any help is much appreciated. All the suggested references, and more, have been added and commented throughout the discussion section of the manuscript.

Methods:

1. A detailed description of when and how sample from baby was collected immediately after birth. Any skin to skin contact with mother? How were the babies kept – separated/with mothers? Were babies washed/cleaned before sampling?

The following sentence was added to the method section: “Samples from neonates were collected immediately after vaginal delivery or caesarean section. Neonates were cleaned by dedicated nurses.

None of them was in skin to skin contact with her/his mother before collecting the nasopharyngeal swab. Neonates were then allowed rooming in with their mothers”

Detailed description of breastmilk samples – what was the hygienic measures employed before collecting sample? Were all samples collected at same time or any time in 10 days – this decreases likelihood of positivity if not collected at closer time to birth as the viral load in maternal circulation will go down anyway.

We added the following sentence in the method section: “According to the WHO, mothers with SARS-CoV-2 infection can breastfeed their babies using appropriate precautions (covering mouth and nose with medical mask, wash hands with soap for at least 20 seconds and after touching the baby, routinely clean and disinfect surfaces have been touched). Breastmilk samples were collected with the same safety measures above”

While in the discussion section: “We decided to collect all breastmilk specimens at five days after delivery (T2) for two reasons: the first one related to a greater production of maternal milk, such as not to interfere with the feeding of the newborn during the first days of life; the second one related to the higher quantity of antibodies in mature milk”

Results:

1. Tables 3 and 4 – is table 4 a summary of results of table 3 – then no need to repeat and could be described easily in text.

The reviewer’s observation on table 4 is correct, indeed it’s a summary of table 3. However, in our opinion, both tables are helpful to monitor the information reported in the manuscript. Table 3 details all the results obtained on the biological samples collected for each patient, it’s extremely informative but could be overwhelming. Conversely, table 4 allows to view at a glance the overall results obtained in the study. The two tables are not interchangeable, as they provide different information, we would prefer to maintain both of them in the final version of the manuscript.

2. Change all indication of “viral” detection to “detection of viral genome” as the test does not dictate virus.

According to the reviewer suggestion, “viral” detection was replaced by “detection of viral genome all along the manuscript.

3. There is no surprising finding that viral genome was detected in vaginal swab – it has been described – PMID 32409520

Thank you for pointing this out. The vast majority of the manuscripts report negative vaginal mucosas and it is very rare to find a SARS-CoV-2 positive ones and we believe it is quite a remarkable finding. However, following your suggestion, we tuned it down and now the sentence reads as follows: “Moreover, we detected the presence of SARS-CoV-2 in vaginal swab...”.

4. Placental injury has also been described in form of maternal vascular malperfusion chronic intervillitis associated with COVID-19 infection. (PMID: 32692408 and several others). How do authors speculate their results are indicative or differentiable from MVM and angiosis versus inflammation?

We would like to thank Reviewer #3 for her/his stimulating suggestion. Unfortunately our data does not allow us to speculate on the proposed topic. As we believe it could produce interesting venues of research, we mentioned it in the bibliography and in the discussion section as follows: “It is reasonable to presume that such inflammatory profile may result in multiple placental malfunctions, as recently reported. However, further experiments are envisaged, in order to confirm this distinctive profile and the consequent pathophysiology.”

Discussion:

1. In light of all the references mentioned above, the discussion will need to be modified to remove all assertions of “first” for anything.

The sentence was modified, according to your suggestion. However, no SARS-CoV-2 positive umbilical cord blood has been reported so far. The opening statement is: “We report for the first time that SARS-CoV-2 is found in the umbilical cord blood. Also, consistently with the literature, we found the virus in the vagina of a pregnant woman and in at-term placentas.”

2. The neonates who are identified to be positive will need to be classified using a proper system of classification with more details on their testing as without which it is difficult to say it is congenital/intrapartum or postpartum transmission. PMID: 32277845

Thank you for the suggestion. Indeed this classification is helpful and likely to be widely used. Therefore, we included such information in the results and in the discussion sections: “...facilitated viral entry through the placenta SARS-CoV-2 positivity of umbilical cord plasma from subject’s 17 newborn proves that infection was acquired antenatally by transplacental transmission as recently established by the international classification on maternal, fetal and neonatal SARS-CoV-2 infection” “SARS-CoV-2 genome was detected in different biological specimens, nonetheless both Vivanti et al. and our study confirm that SARS-CoV-2 transplacental infection occurred, according to Shah et al. classification”. Moreover, such classification was added in table 3, together with other information relevant for the proposed classification.

REVIEWERS' COMMENTS:

Reviewer #1 (Remarks to the Author):

The authors have adequately addressed this reviewer's suggestions.

Reviewer #2 (Remarks to the Author):

Authors did a significant effort to answer all the raised issue and the manuscript is much clearer now. I believe it is adding to our knowledge and worth to be published.

Would only have a clarification about the amniotic fluid samples: were these centrifuged to separate supernatant or analysed w/o centrifugation ? If yes please give the details (g)?

Daniele De Luca

Reviewer #3 (Remarks to the Author):

Thank you for answering many of the questions but still needs some clarifications and corrections.

1. Title should clearly indicate what is the study about. "Evaluation of maternal, placental, neonatal and breast milk samples for viral genome and immunoglobulin from mothers with SARS-COV2 infection during pregnancy". This is the most appropriate title for this report.

2. Still there are places where it is indicated as "We detected the virus in two (6%) maternal plasma samples" on line 221 - these needs to be corrected.; also line 260 "we found the virus in the vagina of a pregnant woman" - again needs to be corrected throughout manuscript.

3. Remove word "strongly" from line 49 of abstract and line 266 in text.

4. Objective 2: "how the production of antibodies occurs in the mother and possibly in the fetus" - you are not studying how they are produced but studying what is their profile.

5. Umbilical cord presence of viral genome is characterized as unique and exclusive for diagnosis by authors - however, they reported that the blood was collected from umbilical vein in response. if that is the case then it still can not be confirmatory of neonatal presence as UV blood still may reflect presence of viral genome in maternal blood and not reflect infection.

6. Discussion needs to be significantly trimmed as it feels like rambling.

7. Figure 1 is not necessary and table 4 can be deleted.

Dear Reviewers,

We are pleased to receive a positive feedback and further forthcoming comments that will definitively improve the quality of our manuscript. We are excited to share our tantalizing results.

Please find below a point-by-point response to each raised issue.

>

> REVIEWERS' COMMENTS:

>

> Reviewer #1 (Remarks to the Author):

>

> The authors have adequately addressed this reviewer's suggestions.

We are pleased to learn that the changes we addressed meet the Reviewer #1 expectations and we are thankful for her/his help.

>

>

> Reviewer #2 (Remarks to the Author):

>

> Authors did a significant effort to answer all the raised issue and the manuscript is much clearer now. I believe it is adding to our knowledge and worth to be published.

We would like to thank the Reviewer #2 for her/his detailed review and the overall positive comments, which led to a significant improvement of our manuscript.

> Would only have a clarification about the amniotic fluid samples: were these centrifuged to separate supernatant or analysed w/o centrifugation ? If yes please give the details (g)?

Amniotic fluid samples were not centrifuged, but analyzed as is.

>

>

> Reviewer #3 (Remarks to the Author):

>

> Thank you for answering many of the questions but still needs some clarifications and corrections.

We would like to thank the Reviewer #3 for her/his in depth review that indeed helped us in addressing many issues.

>

> 1. Title should clearly indicate what is the study about. "Evaluation of maternal, placental, neonatal and breast milk samples for viral genome and immunoglobulin from mothers with SARS-COV2 infection during pregnancy". This is the most appropriate title for this report.

Thank you for the suggestion, which is indeed very appropriate. However, as the Editor suggested a simpler title, we chose a more direct option.

>

> 2. Still there are places where it is indicated as "We detected the virus in two (6%) maternal plasma samples" on line 221 - these needs to be corrected.; also line 260 "we found the virus in the vagina of a pregnant woman" - again needs to be corrected throughout manuscript.

The sentences now read as follow: "We detected SARS-CoV-2 genome in two (6%) maternal plasma samples" and "we found SARS-CoV-2 genome in the vaginal mucosa of a pregnant woman"

>

> 3. Remove word "strongly" from line 49 of abstract and line 266 in text.
Changes have been made accordingly.

>

> 4. Objective 2: "how the production of antibodies occurs in the mother and possibly in the fetus" - you are not studying how they are produced but studying what is their profile.

We agree with the Reviewer #3. However, the final paragraph of the introduction section has been reshaped, following the Editor's suggestion. The new version is: "Here, we report the presence of SARS-CoV-2 genome in the umbilical cord blood and in at-term placentas, in vaginal mucosa of pregnant women and in milk specimen. Furthermore, we report the presence of specific anti-SARS-CoV-2 IgM and IgG antibodies in the umbilical cord blood of pregnant women, as well as in milk specimen. Finally, an intense inflammatory response is triggered by SARS-CoV-2 infection in pregnant women at both systemic and placental level, and, concerning, in umbilical cord blood plasma. Taken together, these results suggest that, although rare, SARS-CoV-2 in-utero vertical transmission is possible, and that the well-known SARS-CoV-2-related inflammatory status is extended to fetuses. Understanding the biological behavior of the virus during pregnancy is essential for defining proper obstetric management of COVID-19 pregnant women."

>

> 5. Umbilical cord presence of viral genome is characterized as unique and exclusive for diagnosis by authors - however, they reported that the blood was collected from umbilical vein in response. if that is the case then it still can not be confirmatory of neonatal presence as UV blood still may reflect presence of viral genome in maternal blood and not reflect infection.

We are pleased to realize that our data triggered some stimulating and constructive scientific discussion. We draw our conclusion based also on the paper of P.S. Shah *et al.* "Classification system and case definition for SARS-CoV-2 infection in pregnant women, fetuses, and neonates.", as strongly recommended by the other Reviewers and utilized by Vivanti *et al.* in the paper "Transplacental transmission of SARS-CoV-2 infection" recently published on Nature Communication. Based on Shah classification, our results indicate a confirmed vertical in-utero transmission, as widely discussed in the manuscript. Moreover, the umbilical cord blood is the fetal blood, flowing only in the fetal circulation and, therefore, a SARS-CoV-2

positive umbilical cord blood will indeed represent the presence of the virus in the fetus itself. Also, the newborn of n. 17, resulted indeed SARS-CoV-2 positive upon delivery and for days. Also, the newborn of n. 17, resulted indeed SARS-CoV-2 positive upon delivery and for days. This observation appears to be consistent with the possibility that the umbilical cord blood was actually positive for the virus itself and not only for the viral genome of maternal origin.

>

> 6. Discussion needs to be significantly trimmed as it feels like rambling.

We are thankful for Reviewer #3 suggestion. Surely, the discussion is not a brief one, but, on the other hand, a 3-page discussion is quite standard. Also, the previous version of the discussion was significantly shorter, but it was expanded following Reviewers #1 and #2 suggestions.

We believe that there are multiple inputs and speculations that are worth to be discussed.

>

> 7. Figure 1 is not necessary and table 4 can be deleted.

We believe that figure 1 would have helped to understand at a glance the timing of samples' collection. However, we are willing to accept Reviewer #3 indication and we removed it.

On the other hand, we believe that table 4 is indeed helpful and it delivers information that are not included in the previous table. While table 3 displays one-by-one all the analyzed parameters for each subject, table 4 displays the percentages, delivering the overall take-home message. Therefore, we believe it is informative and we are prone not to remove it.